# Possible Roles of Transition Metal Cations in the Formation of Interstellar Benzene via Catalytic Acetylene Cyclotrimerization

**DOI:** 10.3390/molecules28217454

**Published:** 2023-11-06

**Authors:** Tatsuhiro Murakami, Naoki Matsumoto, Takashi Fujihara, Toshiyuki Takayanagi

**Affiliations:** 1Department of Chemistry, Saitama University, Shimo-Okubo 255, Sakura-ku, Saitama City 338-8570, Japan; n.matsumoto.503@ms.saitama-u.ac.jp (N.M.); fuji@chem.saitama-u.ac.jp (T.F.); 2Department of Materials & Life Sciences, Faculty of Science & Technology, Sophia University, 7-1 Kioicho, Chiyoda-ku, Tokyo 102-8554, Japan; 3Comprehensive Analysis Center for Science, Saitama University, Shimo-Okubo 255, Sakura-ku, Saitama City 338-8570, Japan

**Keywords:** astrochemistry, interstellar medium, density functional theory, polycyclic aromatic hydrocarbons, transition metal catalysis, cyclotrimerization

## Abstract

Polycyclic aromatic hydrocarbons (PAHs) are ubiquitous interstellar molecules. However, the formation mechanisms of PAHs and even the simplest cyclic aromatic hydrocarbon, benzene, are not yet fully understood. Recently, we reported the statistical and dynamical properties in the reaction mechanism of Fe^+^-catalyzed acetylene cyclotrimerization, whereby three acetylene molecules are directly converted to benzene. In this study, we extended our previous work and explored the possible role of the complex of other 3d transition metal cations, TM^+^ (TM = Sc, Ti, Mn, Co, and Ni), as a catalyst in acetylene cyclotrimerization. Potential energy profiles for bare TM^+^-catalyst (TM = Sc and Ti), for TM^+^NC^−^-catalyst (TM = Sc, Ti, Mn, Co, and Ni), and for TM^+^-(H_2_O)_8_-catalyst (TM = Sc and Ti) systems were obtained using quantum chemistry calculations, including the density functional theory levels. The calculation results show that the scandium and titanium cations act as efficient catalysts in acetylene cyclotrimerization and that reactants, which contain an isolated acetylene and (C_2_H_2_)_2_ bound to a bare (ligated) TM cation (TM = Sc and Ti), can be converted into a benzene–metal–cation product complex without an entrance barrier. We found that the number of electrons in the 3d orbitals of the transition metal cation significantly contributes to the catalytic efficiency in the acetylene cyclotrimerization process. On-the-fly Born–Oppenheimer molecular dynamics (BOMD) simulations of the Ti^+^-NC^−^ and Ti^+^-(H_2_O)_8_ complexes were also performed to comprehensively understand the nuclear dynamics of the reactions. The computational results suggest that interstellar benzene can be produced via acetylene cyclotrimerization reactions catalyzed by transition metal cation complexes.

## 1. Introduction

Polycyclic aromatic hydrocarbons (PAHs) play an important role in regulating the physical and chemical conditions of interstellar medium. Moreover, they are possible sources of unidentified infrared emission bands (UIBs) and diffuse interstellar bands (DIBs) [1,2,3,4,5,6,7]. Exploring the mechanisms of PAH formation may clarify their abundance and presence in space. Understanding the mechanism underlying the formation of benzene, the simplest building block of PAHs with an aromatic ring structure, from small precursor molecules under interstellar conditions is also important [1,2,3]. However, it is generally difficult to understand the formation mechanism of interstellar benzene directly from astrochemical observations. Several scenarios have been previously proposed. For example, Woods et al. [8] proposed a stepwise ion–molecule reaction mechanism involving the reactions C_4_H_3_^+^ + C_2_H_2_ → *c*-C_6_H_5_^+^ + *hν* and *c*-C_6_H_5_^+^ + H_2_ → *c*-C_6_H_7_^+^ + *hn*, followed by dissociative electron attachment: *c*-C_6_H_7_^+^ + e^−^ → C_6_H_6_ + H. The validity of this mechanism was examined by comparing the benzene column density observed in CRL 618 with calculations using the chemical evolution model. Kaiser et al. [9] studied the neutral C_2_H + C_4_H_6_ (1,3-butadiene) → C_6_H_6_ + H reaction in the gas phase using sophisticated crossed molecular beam experiments. This neutral bimolecular reaction is barrier-less and highly exothermic, leading to the presumption that it can proceed efficiently in cold molecular clouds. Alternatively, El-Shall et al. [10,11] suggested that the benzene radical cation is formed via the ionization and subsequent polymerization of large acetylene clusters, (C_2_H_2_)*_n_*, and that these processes may be responsible for the formation of benzene and other PAHs in space. Zhao et al. [12] experimentally showed that the catalytic conversion reactions of acetylene on the surface of a solid SiC grain efficiently form benzene and PAHs and emphasized the importance of the dangling bonds of the SiC solid in the catalytic p-bond breaking of acetylene.

The direct synthesis of benzene from three acetylene molecules is generally referred to as [2 + 2 + 2] acetylene cyclotrimerization. Due to the large activation barrier associated with the cleavage of π-bonds, [13,14] non-catalytic cyclotrimerization is very slow even at high temperatures [15]. The formation mechanism of metal-catalyzed acetylene cyclotrimerization, therefore, was extensively studied in the field of organic chemistry [16,17]. Acetylene complexes with transition metal ions as catalysts were studiously investigated by Duncan and coworkers [18,19,20,21,22,23,24,25]. They observed the infrared spectra of acetylene complexes with several transition metal ions, including iron, [18] nickel, [19,20] copper, [21] silver, [22] gold, [23] zinc, [24] and vanadium [25]. The spectroscopic data provided evidence of cyclotrimerization of the V^+^(C_2_H_2_)_3_ complexes to form V^+^(benzene) via metallacycle intermediates [25]. Moreover, Duncan and coworkers showed that there is a large activation barrier for the cyclotrimerization reaction [18]. The activation barrier becomes the key limitation in the cyclotrimerization process under the interstellar condition.

Through density functional theory (DFT) calculations of the quantum chemical reaction paths, we previously demonstrated that the conversion of the (C_2_H_2_)_3_-Fe^+^-L complex to the C_6_H_6_-Fe^+^-L complex proceeds via transition states, with relatively lower barriers compared to those in the cyclotrimerization of the complexes with the bare iron ion [26,27,28]. It was found that L = (H_2_O)*_n_* (*n* = 8, 10, 12, and 18) clusters as the ligand reduced the number of transition states [26,28]. We also studied the cyclotrimerization of (three) acetylene units catalyzed by Fe^+^ with several ligands, namely, F^−^, Cl^−^, OH^−^, SH^−^, NC^−^, NH_3_, H_2_O, cyclopentadienyl anion (Cp^−^), CN^−^, benzene (Bz), and PH_3_ [27]. The Fe^+^-catalyzed cyclotrimerization using three of these ligands (CN^−^, Bz, and PH_3_) had more than two additional transition state barriers, whereas the former eight ligand complexes formed benzene through one transition state barrier. Moreover, in dynamics simulations, 44 trajectories out of 50 generated benzene forms within 1 ps with the NC^−^ ligand [27]. From these results, we tentatively speculate that the TM^+^-(H_2_O)*_n_* and TM^+^-NC^−^ compounds may act as an effective catalysis for forming benzene via acetylene cyclotrimerization.

Iron is the sixth most abundant element in astrophysical environments [29]. However, the role of transition metals is not yet understood from an astrochemical viewpoint [30,31]. Herein, we extended our previous work on Fe^+^ to earlier 3d transition metal cations, namely Sc^+^ and Ti^+^, and later ones, which are Mn^+^, Co^+^, and Ni^+^. Notably, atomic metals and their cations, including Ti and Mn, have been previously detected in the circumstellar envelope of the asymptotic giant branch carbon star IRC+10216 [32]. Furthermore, TiO and TiO_2_ have been detected in the gas phase through rotational transitions at submillimeter wavelengths toward the red supergiant VY CMa [33]. The interstellar spectra of ζ Ophiuchi have been observed, where the lines of cobalt and nickel ions were detected [34,35]. Scandium has been studied as a key element of Am star due to the deficiency of Sc [36]. Nevertheless, weak lines of scandium were also detected in Am star [37]. Although the Sc^+^-ligand complex might not be effective for cyclotrimerization in the interstellar medium due to the Sc deficiency, investigating the reaction mechanism using scandium, which is the earliest in the 3d transition metal series, may be important for understanding the cyclotrimerization mechanisms involving 3d transition metals from a theoretical viewpoint.

In this study, using quantum chemistry calculations and reaction dynamics simulations, we investigate the catalytic efficiency of compounds composed of the aforementioned transition metals in acetylene cyclotrimerization reactions leading to benzene-metal complexes. The results are discussed in Section 2. In the first place, the potential energy profiles of the bare Sc^+^ and Ti^+^ catalyst systems without any ligand for three acetylene cyclotrimerization were investigated and are discussed in Section 2.1. Early transition metals such as scandium and titanium, which have more unoccupied *d* orbitals than Mn^+^, Co^+^, and Ni^+^, are the focus in the subsection. We discuss the barrier-reduction effects of bare Sc and Ti ions. Next, the reaction mechanisms for the complexes containing TM^+^NC^−^ compounds (TM = Sc and Ti) are discussed and compared with the bare TM catalyst system cases in Section 2.2. In the previous study, the gas-phase FeCN and MgCN/MgNC molecule was detected by observing the rotational spectra toward the envelope of IRC+10216 [38,39,40,41]. It was also expected that the NC^−^ ligand would act as an effective catalysis from our previous theoretical study [27]. We also calculated the potential energy profiles for the acetylene complexes with the TM^+^NC^−^ ligands (TM = Mn, Co, and Ni) and compared the barrier height with those of ScNC and TiNC catalyst systems in the same subsection. Thirdly, the potential energy profiles of the Sc^+^ and Ti^+^ catalysts with the (H_2_O)_8_ cluster as a ligand are discussed in Section 2.3. Solid water (ice) is known to play a very important role in interstellar surface reactions [42]. As in our previous theoretical study, [26] the (H_2_O)_8_ cluster was utilized. The previous study showed that the cubic structure with a highly symmetric *D*_2*d*_ character is the most stable for the (H_2_O)_8_ cluster. Thus, the effects of the various low-lying structures can be neglected because of the complicated hydrogen-bonding motif. Note that the (H_2_O)_8_ cluster with bare transition metal ions might not retain the cubic structure because of the large exothermic energy due to the relatively strong attractive interaction between the ice cluster and the bare metal ions. However, if complex transition metal systems such as TM^+^-NC^−^ and TM^+^-H_2_O catalysts are adsorbed onto the ice cluster, the exothermic interaction energies could be reduced, indicating that the (H_2_O)_8_ cluster and larger clusters can possibly maintain the cubic structure without forming a solvated metal ion structure. In the study of a catalysis reaction on interstellar ice, the complex structures of MgCN with 17H_2_O and 24H_2_O were investigated, as well as the structure of the ice cluster [43]. Therefore, the (H_2_O)_8_ cluster was been selected as the ligands in this study. In the last part of Section 2, the results of reaction dynamics simulations in terms of the acetylene cyclotrimerization for Ti^+^NC^−^ and Ti^+^(H_2_O)_8_ catalyst systems are discussed. The potential energy profiles through the intrinsic reaction coordinate (IRC) using quantum chemistry calculations effectively provide statistical information such as the properties of stational points to understand the reaction mechanism. The reaction dynamics, however, do not often follow the IRC [44,45]. Non-IRC dynamics play a significant role in many organic chemistry reactions, including catalytic reactions. Our previous study in terms of Fe^+^-catalyzed acetylene cyclotrimerization presented the fact that many trajectories exhibited non-IRC dynamics [27]. The performing of molecular dynamics simulations could be of critical importance to understand the dynamic properties of the chemical reactions. Motivated by this, we performed the molecular dynamics simulations in this study. Finally, a summary and future directions of this study are presented in Section 3. The computational details of the quantum chemistry calculations and molecular dynamics simulations are presented in the last part of this paper.

## 2. Results and Discussion

### 2.1. Bare Sc^+^ and Ti^+^ Catalyst Systems

Figure 1a,b shows the potential energy profiles (obtained from the B3LYP-D3(BJ)/def2-TZVP calculations) of the acetylene cyclotrimerization reactions for the conversion of the reactant (C_2_H_2_ + (C_2_H_2_)_2_-TM^+^) to the product complex (PC) (C_6_H_6_-TM^+^) catalyzed by the TM^+^ (TM = Sc and Ti) complexes. The horizontal axis represents the reaction path length measured within the mass-weighted coordinate system, and the origin of the corresponding axis is taken to be the first intermediate complex, which is primarily produced by the association of the (C_2_H_2_)_2_-TM^+^ reactant with an additional C_2_H_2_ molecule. Zero energy is defined as the energy level of the C_2_H_2_ + (C_2_H_2_)_2_TM^+^ reactants, and no zero-point energy correction was included (the harmonic zero-point energy values for all stationary points found in this study are reported in the Appendix A). Figure 1a,b clearly shows that both reactions from the reactants to PCs are totally exothermic. Further details such as the molecular structures at the stationary points and the reaction and the reaction paths are discussed later. The potential energies at each stationary point from the reactants to products (benzene + TM^+^) via the intermediate (INT1) and transition (TS1) states and their vertical excitation energies are shown in Table 1. The spin state *i* of the reactants corresponds to the superscript of *^i^*{^1^C_2_H_2_, *^i^*Sc^+^(C_2_H_2_) (*i* = 1, 3, and 5)} and *^i^*{^1^C_2_H_2_, *^i^*Ti^+^(C_2_H_2_) (*i*2, 4, and 6)}. The spin states of the products are represented as *^i^*{^1^benzene, *^i^*Sc^+^(C_2_H_2_) (*i*1, 3, and 5)} and *^i^*{^1^benzene, *^i^*Ti^+^ (*i* = 2, 4, and 6)}, respectively. From the reactants to PCs, the potential energies of the lowest spin state were found to be consistently lower than that of the higher spin state. For example, in the case of Sc^+^, the spin state of the electronic ground state of the isolated Sc^+^ cation is known to be a triplet; however, the spin states of the (C_2_H_2_)_3_Sc^+^ in the electronic ground state are singlets. Similarly for Ti^+^, the potential energy surface of the doublet was lower than that of the quartet and sextet. The Sc- and Ti-catalyzed product fragments have triplet and quartet electronic ground states, as expected. The highest spin states for both systems do not seem to contribute to the cyclotrimerization because their potential energies through the overall reaction pathway relatively represent large values. The quintet and sextet energies of the products are extremely high because the electron configuration of isolated Sc^+^ is [Ar]3d^1^4s^1^ and that of Ti^+^ is [Ar]3d^2^4s^1^, and excitation from the inner shell is required. Herein, an affordable pathway to obtain the final product fragments should be emphasized. Under low-density conditions, the intramolecular redistribution of vibrational energies derived from the exothermic energy could play an important role in releasing the transition metal cation. Furthermore, one of the most reasonable pathways to obtain the final product fragments is the dissociative electron attachment: (benzene-TM^+^) + e^−^ → benzene + TM [46,47]. For example, the reaction involving a spin transition, (^1^benzene-^1^Ti) → ^1^benzene + ^3^Ti, is only a 0.5 kcal/mol endothermic reaction (see Appendix A). The vibrational continuum and resonance states of the neutral product complex involved in the electron attachment provide relatively large vibrational energies to the complex. The metal–molecule complex therefore enables the dissociation to the benzene and neutral metal fragments. Alternative pathways are photodissociation by irradiation of the cosmic rays and collisional dissociation by encountering other systems in the dense clouds [46,47].

Figure 2a,c shows the optimized geometries and their Kohn–Sham molecular orbitals (MOs) for the reactant fragments TM^+^(C_2_H_2_)_2_ (TM = ^1^Sc and ^2^Ti). We also optimized ^3^Sc^+^(C_2_H_2_)_2_ and ^4^Ti^+^(C_2_H_2_)_2_, which are shown in Figure 2b,d, with their MOs. It is known that the formation of a metallacycle between a transition metal and acetylene (alkyne) is affected by the electron donor–acceptor interactions between the *d* orbitals of the metal and π-orbitals of the acetylene [48]. Two π-orbitals, which are parallel (π_∥_) and perpendicular (π_⊥_) to the TMCC plane, respectively, interact with the unoccupied d-orbitals of the metal. Moreover, the d-donor orbital interacts with the π_∥_* orbital, which is known as π-back-donation. This metal–olefin bonding belongs to the Dewar–Chatt–Duncanson model [49]. In this study, the number of electrons in the d-donor orbital could be two and three for Sc^+^ and Ti^+^, respectively, from the electron configurations of isolated Sc^+^ and Ti^+^. As shown in Figure 2a, the 24 α and 25 α orbitals of the singlet state, which are the highest occupied molecular orbital (HOMO) and lowest unoccupied molecular orbital (LUMO), respectively, contribute to π-back-donation. Thus, one acetylene could only be influenced by the d-π_∥_* interaction. The metallacycle is formed by d-π_∥_ and d-π_⊥_ interactions only on the other side. The MOs correlated with the d-π_∥_ and d-π_⊥_ interactions are shown in Appendix A. The C_2_H_2_ moiety on the right side of the Sc cation was influenced by the back-donation from 24 α orbital. The C-C bond was elongated to 1.33 Å and the C-C-H bond angle was distorted to 131.5° from the equilibrium structure of the isolated singlet C_2_H_2_ (the C-C bond = 1.20 Å and the C-C-H angle = 180.0° [50]). The coordinated structure was quite similar to the triplet C_2_H_2_ (the C-C bond = 1.32 Å and the C-C-H angle = 128.3° [50]). It is represented that the triplet configuration of acetylene plays quite a significant role in the π-donation and π-back-donation process between acetylene and the 3d metal. The result is quite consistent with the chemisorption of the acetylene on the copper surface [50]. On the other hand, the C_2_H_2_ part on the left side had the singlet configuration, which indicates that the occupied π-orbital of the singlet C_2_H_2_ donates directly to the unoccupied metal d-orbital. Moreover, the Sc-C bond distance (2.66 Å) was larger than the right side one (2.02 Å). The π-back-donation is significantly correlated with the 3d-metal-acetylene coordination. From these acetylene structures, the nature of the bond formation between the Sc cation and acetylene based on the donor and acceptor processes of π-electrons follows a spin-uncoupling mechanism, [50,51,52] which is the coordination process whereby unpaired electron configurations for both metal and unsaturated hydrocarbons interact with each other. In contrast with the singlet Sc^+^(C_2_H_2_)_2_, the singly occupied highest and second highest molecular orbitals (SOMO and SOMO-1) of triplet Sc^+^(C_2_H_2_)_2_ are the 25 α and 24 α orbitals, as shown in Figure 2b. The two half-occupied orbitals symmetrically contribute to π-back-donation and then the molecular structure became *C_2v_*. Both C-C bond lengths (1.24 Å) are longer than the isolated singlet configuration and shorter than the triplet one. In comparison with the singlet Sc^+^(C_2_H_2_)_2_, the Sc-C bond distances are longer than the length of the C_2_H_2_ moiety under the strong influence of back-donation and shorter than the bond distance of the acetylene part, which the back-donation does not affect. It is suggested that π-back-donation plays an important role in the spin-uncoupling process and the acetylene conformation change. Note that the potential energy of the triplet Sc^+^(C_2_H_2_)_2_ was 9.7 kcal/mol, which is higher than the one of the reactant fragment. Figure 2c,d shows the doublet (0.0 kcal/mol) and quartet (16.0 kcal/mol) Ti^+^(C_2_H_2_)_2_ equilibrium structures and the Kohn–Sham molecular orbitals at their minima. The 25 α (SOMO) and 24 α (HOMO) were assigned as the d-π_∥_* orbitals for the doublet state, whereas the quartet-state compound has three back-donation orbitals, namely the 26 α (SOMO), 25 α (SOMO-1), and 24 α (SOMO-2) orbitals. The acetylene configurations of both doublet and quartet Ti^+^(C_2_H_2_)_2_ compounds are influenced by the π-back-donation from the occupied 3d-orbitals of Ti cation, which is similar to the case of Sc-catalyst systems. The Ti-C bond distances are relatively shorter than the related Sc-C lengths. Especially, the Ti-C bond distance (2.28 Å) on the left side of Ti shown in Figure 2c was extremely shorter than the one (2.66 Å) of Sc shown in Figure 2a because the 25 α (SOMO) of doublet Ti^+^(C_2_H_2_)_2_ contributed to the Ti-acetylene-coordination, while the related orbital character for singlet Sc^+^(C_2_H_2_)_2_ belonged to the 25 α (LUMO). The back-donation from metal cations is therefore of critical importance for the coordination. These results are quite consistent with the relation between the metal-ligand distance and back-donation in the Dewar–Chatt–Duncanson model [53].

As shown in Figure 1a,b, the TM^+^C_4_H_4_ five-membered metallacycle structures (INT1) were formed without potential barriers, which was confirmed through the geometry optimization (See Appendix A). It was predicted that the occupied d-donor orbitals were insufficient to maintain the forms of the three acetylenes for the ^1^Sc^+^- and ^2^Ti^+^-catalyst systems. Figure 3 shows molecular structures at optimization steps 10, 20, and 52 for the ^2^Ti^+^-catalyst system (See Appendix A) and the crucial molecular orbitals. In 31 α (HOMO) at step 10, one d-donor orbital is shared with C1-C3 π* and C2-C4 π* orbitals of each acetylene, where the C1-C2 length is 2.03 Å. The 29 α (HOMO-2) at step 10 is due to the π-donation to the unoccupied metal d-orbital from the occupied C1-C3 π and C2-C4 π orbitals. The Ti-C1 and Ti-C2 coordination interactions are correlated with 33 α (LUMO). During the optimization process to step 20, the C1-C2 bond (1.52 Å) was generated with the C1-C2 σ-orbital in 29 α (HOMO-2). This MO was stabilized (31 α → 29 α) and the MO included a node between C1 and C2 was destabilized (29 α → 30 α) from steps 10 to 20. The Ti-C1 (Ti-C2) bond was elongated from 2.18 (2.27) Å to 2.28 (2.31) Å with the 35 α (LUMO+2) which does not interact between the 3d and C1-C3 (C2-C4) π orbitals. Eventually, a five-membered metallacycle intermediate of the doublet Ti^+^(C_4_H_4_)C_2_H_2_ was generated in step 52. This metallacycle formation from the reactants is similar to the titanium metallacyclobutane formation from an ethylene and titanium methylidene, which is the barrierless reaction [54]. The π-back-donation does not contribute to the reaction. As mentioned above, the coordination conformations between the metal and unsaturated hydrocarbon are critically influenced by the π-back-donation based on the spin-uncoupling mechanism. In the cases of the ^2^Ti^+^-catalyst as well as ^1^Sc^+^-catalyst systems, the d-donor electrons contributing to the π-back-donation are not enough to keep three acetylenes separately. It is suggested that a small number of occupied d-donor orbitals induce the barrierless formation of the five-membered metallacycle.

Figure 1a,b shows that TM^+^-benzene (PC) is produced through a relatively low transition state (TS1) from the 3/5 dimetallacycle (INT1). The IRC profiles are found to be similar to the pathway to the vanadium cation ([Ar]3d^3^4s^1^) case [25]. The barrier heights of TS1 are 2.4 and 7.4 kcal/mol for the Sc^+^ and Ti^+^ ions, respectively. The π-back-donation is not correlated with the Sc^+^C_2_H_2_ coordination bond, whereas the formation of Ti^+^C_2_H_2_ is influenced by the d-π_∥_* interaction involving one SOMO. A weaker Sc^+^C_2_H_2_ coordination could relatively reduce this barrier. Linares and coworker presented that the metal–ligand distance becomes more lenient in the absence of back-donation in the Dewar–Chatt–Duncanson model [53]. Our work quite agrees with the previous study. As mentioned above, it is found that this three acetylene cyclotrimerization with the TM^+^ catalysis is a totally exothermic reaction.

### 2.2. TM^+^NC^−^ Catalyst System

Figure 4 shows the potential energy profiles (obtained from B3LYP-D3(BJ)/def2-TZVP calculations) of the acetylene cyclotrimerization reactions (from the C_2_H_2_ + (C_2_H_2_)_2_-TM-NC reactant to the C_6_H_6_-TM-NC product complex) catalyzed by the TM^+^NC^−^ (TM = Sc and Ti) compounds. The lowest electronic states in the overall reaction pathways to the PCs for ScNC and TiNC catalyst systems are the singlet and doublet state, respectively (See Appendix A), and the benchmark results are shown in Appendix A. As shown in Figure 4a, the two acetylene molecules of the (C_2_H_2_)_2_-ScNC reactant complex spontaneously dimerized to form a Sc-C_4_H_4_-type five-membered metallacycle structure without an entrance barrier. This was confirmed by the geometry optimization (see Appendix A). A second transition state (TS2) was also observed in the reaction pathway from the Sc^+^C_6_H_6_ seven-membered metallacycle intermediate (INT2) to the benzene product complex (PC), although the corresponding barrier height measured for this intermediate was extremely small (0.1 kcal/mol). This suggests that the seven-membered metallacycle intermediate (INT2) is located in a shallow potential energy well. Another interesting result was observed for the TiNCs, as shown in Figure 4b. In this case, the Ti-C_4_H_4_ five-membered metalacyclic structure was not observed in the initial (C_2_H_2_)_2_-TiNC reactant; however, there was no barrier between the C_2_H_2_ + (C_2_H_2_)_2_-TiNC reactants and the C_2_H_2_-TiNC-C_4_H_4_ intermediate complex. Thus, the association of the third acetylene group with (C_2_H_2_)_2_-TiNC to form the C_2_H_2_-TiNC-C_4_H_4_ intermediate complex (INT1) is barrierless. This behavior was confirmed by geometric optimization (see Appendix A), minimum energy path calculations (see Appendix A), and BOMD calculations (see below). The INT1 connects to the seven-membered metallacycle intermediate (INT2) via a transition state (TS1), as shown in Figure 4b, while there is no second intermediate in the bare Ti^+^ system, as shown in Figure 1b. One possible reason for the existence of INT2 is that the interactions between the *d* orbital of the titanium cation and the occupied nonbonding two π and σ* orbitals of NC^−^ may stabilize the potential energy of the ring-opening C_6_H_6_ compound. Further details are out of scope in this study because the TS2 barrier height was extremely small (0.6 kcal/mol), which indicates that the INT2 is located in a very shallow potential energy well. Eventually, another transition state (TS2) connects the INT2 to the TiNC-benzene PC. The overall reaction process including these two transition states is found to be highly exothermic.

Figure 5 shows the difference in the barrier height of the bare metal versus the TMNC catalysts. The potential energy of the intermediate complex (INT1) was set to 0 kcal/mol. The zero point on the horizontal axis representing the reaction path length was set at the TS1 structure. In the case of scandium, the NC^−^-ligated complex has a slightly higher energy (0.5 kcal/mol) than that of the bare metal compounds shown in Figure 5a. In contrast to the ^1^Sc^+^ complexes, the barrier height at TS1 with bare titanium (7.4 kcal/mol) was significantly decreased by the NC^−^ ligand. The potential energy of the TiNC compound in TS1 was 2.2 kcal/mol. Figure 6 shows the imaginary frequencies at the TS1. While the frequencies for the Sc^+^ (215*i* cm^−1^) and ScNC (266*i* cm^−1^) catalyst systems are similar, the frequency of the TiNC system (72*i* cm^−1^) is quite a bit smaller than the one of the Ti^+^ system (269*i* cm^−1^), which indicates that the potential curvature along the reaction coordinate around TS1 for the TiNC system is quite lenient compared with the curvature for Ti^+^ catalyst system. This behavior leads to a reduction in barrier height.

The potential energy profiles calculated by the B3LYP-D3(BJ)/def2-SVPP level for the MnNC, CoNC, and NiNC catalyst systems are shown in Figure 7. In contrast with the Sc+ and Ti+ systems, the Mn^+^, Co^+^, and Ni^+^ systems have a stable structure corresponding to the TM-NC-(C_2_H_2_)_3_ structure, in which the three acetylene molecules are coordinated to the transition metal cation center. In addition, there is a substantial barrier between the TM and NC–(C_2_H_2_)_3_ structure and the C_2_H_2_-TM-NC-C_4_H_4_ intermediate complex, as shown in Figure 7 In particular, for Co^+^ and Ni^+^, the energy levels of these barriers are higher than the energy levels of the reactant. The results suggest that the catalytic formation of benzene from the three acetylene molecules is energetically favored. However, one can easily understand that the ScNC and TiNC compounds can act as efficient catalysts in the cyclotrimerization of acetylene because there is no substantial barrier along the reaction pathways. The catalytic efficiency of the Sc and Ti compounds in the acetylene cyclotrimerization process can be qualitatively understood by considering the number of electrons in the 3d orbitals of the transition metal cation. Three unoccupied 3d orbitals of the metal cation are required to interact strongly with three π-orbitals of the three acetylene molecules, which can form three 3d-π bonding orbitals. Note that each acetylene molecule has two π-orbitals, which may account for the absence of a large barrier in the cyclotrimerization pathways from acetylene to benzene for the ScNC and TiNC catalysts. Therefore, we investigated the catalytic efficiencies of the Sc^+^(H_2_O)*_n_* and Ti^+^(H_2_O)*_n_* clusters in acetylene cyclotrimerization.

### 2.3. TM^+^(H_2_O)_8_ Catalyst System

As a tentative model catalyst system, we employed the TM^+^(H_2_O)_8_-ligand to form a cubic cluster structure throughout the reaction-path calculations in this study. The potential energies were calculated at the B3LYP-D3(BJ)/def2-SVPP level. Figure 8a shows the reaction pathways for acetylene cyclotrimerization catalyzed by the Sc^+^(H_2_O)_8_ complex. The spin state of the lowest potential energy surface was a singlet, similar to those of bare Sc^+^ and ScNC. The scandium cation was preferentially bound to the O atom of water with no dangling OH bonds. This is similar to the binding in the Fe^+^ system [26]. Moreover, clearly the Sc-C_4_H_4_ five-membered metallacycle structure was already formed in the reactant complex. This is similar to the structure of ScNC. There is only one transition state in the reaction pathway from the intermediate complex to the benzene complex, the barrier height of which is 3.9 kcal/mol, as measured from the intermediate energy level, and is slightly higher than that for the ScNC system (2.3 kcal/mol of the def2-SVPP level, see Appendix A). From this figure, it is qualitatively deduced that the association of the third acetylene group with the C_4_H_4_-Sc^+^(H_2_O)_8_ complex spontaneously produces the benzene-Sc^+^(H_2_O)_8_ complex. Figure 8b shows similar results for the Ti^+^(H_2_O)_8_ complexes. The barrierless reaction was confirmed by MEP calculation (see Appendix A). The reaction pathway is similar to that in the case of TiNC.

### 2.4. Molecular Dynamics Simulations

Although the static reaction path calculations presented in Figure 4b and Figure 8b show that TiNC and Ti^+^(H_2_O)_8_ can serve as efficient catalysts for benzene formation via acetylene trimerization, it would be interesting to understand these reaction processes from a dynamical perspective. To this end, BOMD calculations were performed. The BOMD calculations were initiated with the optimized structures of both the TiNC-(C_2_H_2_)_2_ (or Ti^+^(H_2_O)_8_-(C_2_H_2_)_2_) reactant complex and C_2_H_2_, where the third acetylene molecule is initially located in such a manner that the distance between Ti and the midpoint of C≡C is 4–5 Å, with a random orientation. Initially, no kinetic energy was applied to the atoms in the system for qualitative simulation of the reaction at very low temperatures under interstellar conditions. However, these atoms can obtain kinetic energy during the trajectory evolution owing to the large attractive force (~50 kcal/mol, see Figure 4b and Figure 8b) between C_2_H_2_ and the Ti^+^ moiety in the complex. These initial conditions are physically unrealistic; however, such computational conditions are reasonable for understanding the reaction pathways of benzene formation from a dynamic perspective. Six and five BOMD trajectories were integrated for the TiNC and Ti^+^(H_2_O)_8_ cases, respectively.

Figure 9a shows a typical BOMD trajectory (trajectory #1) for the TiNC complex, where the potential energy is plotted as a function of the simulation time, along with a few selected structures. In the early stage of this trajectory (*t* < 600 fs), the third acetylene molecule approaches the Ti^+^ moiety in the complex because of attractive forces, as previously mentioned. At *t* ≈ 700 fs, the two carbon atoms in the two acetylenes approach each other, forming a CC bond, thereby generating a five-membered metallacycle intermediate with a C_2_H_2_-TiNC-C_4_H_4_ structure. This intermediate structure was maintained up to *t* = 1500 fs. At *t* ≈ 1500 fs, a new CC bond is formed between C_2_H_2_ and the C_4_H_4_ moiety, producing a seven-membered metallacycle intermediate with a TiNC-C_6_H_6_ structure. The lifetime of this intermediate is not long (~300 fs), and the benzene structure is readily formed at *t* ≈ 1900 fs in this trajectory. These real-time nuclear dynamic behaviors are qualitatively consistent with the potential energy profile of the reaction path presented in Figure 4b. The results for the other four BOMD trajectories are shown in Figure 9b. In trajectories of the #2–#5 cases, benzene was finally formed within a simulation time of 2.5 ps, although the lifetime of the intermediate complex differed slightly depending on the nature of the BOMD trajectories. We calculated another trajectory (Trajectory #6 in Figure 9b) in which the third acetylene molecule was initially placed on the other side of the Ti moiety of the TiNC–(C_2_H_2_)_2_ complex (see the inset structure in Figure 9b). Interestingly, the third acetylene molecule does not approach Ti^+^ because of the presence of the other two C_2_H_2_ molecules. In this case, the trajectory consistently bounced dynamically between the third acetylene and the two acetylenes bound to Ti^+^ during the simulation. This suggests that the initial position of the third acetylene molecule plays an important role in the dynamics of benzene formation. Appendix A shows the initial configuration of the TiNC–(C_2_H_2_)_2_ complex with the isolated acetylene for all trajectories. At *t*0 fs, the acetylene was located below the CCCC plane of the two acetylene molecules bound to Ti^+^ and there was no obstacle between the isolated acetylene and Ti^+^ in Trajectories #1–#5. As mentioned above, the bond formation between Ti^+^ and acetylene in Trajectory #6 was prevented by the two acetylenes. The benzene formation movie is presented as Appendix A.

Figure 10a shows a typical benzene-forming BOMD trajectory (Trajectory #1) calculated for the Ti^+^(H_2_O)_8_ complex, where the potential energy is plotted as a function of time, along with selected pictures of the geometry. In this trajectory, the C_2_H_2_-Ti-C_4_H_4_ five-membered metallacycle intermediate was formed in the early stages (after *t* ≈ 200 fs). This intermediate structure was maintained up to *t* ≈ 500 fs and a CC bond was formed between the C_2_H_2_ and the C_4_H_4_ moieties in this intermediate complex to generate a TiNC-C_6_H_6_ seven-membered metallacycle structure at *t* ≈ 850 fs, followed by benzene formation. Thus, a benzene complex is formed within a short period (1 ps) in this trajectory. Interestingly, proton transfer from the water molecule directly attached to the Ti^+^ cation to another water molecule occurs at *t* ≈ 580 fs. Thus, after this time, the structure of the Ti^+^(H_2_O)_8_ complex approximately assumes the form of Ti^+^OH^−^·(H_2_O)_5_·(H_5_O_2_^+^) or Ti^+^OH^−^·(H_2_O)_6_·(H_3_O^+^), where the H_5_O_2_^+^ and H_3_O^+^ structures are generally called Zundel and Eigen ions, respectively [55]. This is very interesting because the calculation of the reaction path presented in Figure 8b does not indicate any proton transfer along the reaction pathway. In this regard, the trajectory presented in Figure 10a shows a non-IRC dynamics feature [44,45]. The occurrence of proton transfer suggests that the OH bond in water, which is directly bound to Ti^+^, is somewhat weak in the complex. This behavior is consistent with the fact that the corresponding OH bond lengths for all the stationary points on the potential energy surface were slightly longer (1.05–1.08 Å, see Figure 8b) than those for an isolated neutral water molecule (~0.96 Å). Thus, energy transfer can also occur from the (C_2_H_2_)_3_-Ti^+^ moiety to the water clusters through proton transfer during the dynamic process, although the amount of energy transferred is not very large because of the light nature of protons. This can be understood from evaluation of the water cluster dynamics after formation of the benzene complex. Although the hydrogen-bonded structure was deformed to a certain degree at this stage, water evaporation was not observed within the simulation time. Thus, the nuclear dynamics are in contrast to those of a simple TiNC complex. Benzene formation movies from the two viewpoint angles are presented as Appendix A.

The results obtained with the other four BOMD trajectories are shown in Figure 10b. Three of these trajectories led to the formation of the benzene complex, although the formation time depended on the nature of the trajectory. However, the benzene complex was formed later (2–3 ps) for these trajectories compared to benzene formation in Trajectory #1 (Figure 10a). Thus, the process of generating the benzene complex (within 1 ps), as shown in Figure 10a, was remarkably fast. The slow production of the benzene complex presented in Figure 10b contrasts with the trajectories calculated for the TiNC complex; however, the results are consistent with the potential energy profile along the reaction path presented in Figure 9b, where the barrier height from the C_2_H_2_-Ti-C_4_H_4_ five-membered metallacycle intermediate is 5.2 kcal/mol. This barrier height was slightly larger than that of the TiNC complex (2.0 kcal/mol, see Appendix A). Moreover, the number of nuclear degrees of freedom for the Ti^+^(H_2_O)_8_ system is higher than that for the TiNC system, which may affect the relaxation of the kinetic behavior. Thus, it is reasonable to assume that the lifetimes of the five- or seven-membered metallacycle intermediates in the Ti^+^(H_2_O)_8_ system were longer than those in the TiNC system. To estimate the average benzene formation time, several BOMD trajectories with longer simulation times were integrated to obtain statistically meaningful outcomes. However, these calculations are beyond the scope of this study.

Trajectory #3 does not lead to benzene formation up to *t* = 3.5 ps but leads to a seven-membered metallacycle intermediate structure. Interestingly, the geometric structure at *t* = 3–3.5 ps was nearly planar (Figure 10b). Trajectory #3 also shows non-IRC dynamics because such a planar structure cannot be observed along the IRC pathway presented in Figure 8b. Finally, proton transfer in a water molecule directly bound to Ti^+^ was observed in all trajectories. A detailed analysis indicated rearrangement of the hydrogen bonding in the water molecule directly bound to Ti^+^. Initially, this water molecule is hydrogen bonded to two other water molecules by donating two hydrogen atoms (see Figure 8b). During trajectory evolution, the structure of the cluster is rearranged such that the corresponding water is hydrogen bonded to one water molecule.

## 3. Conclusions and Future Directions

In this study, we first performed reaction path analyses using quantum chemistry calculations of the acetylene cyclotrimerization reaction catalyzed by transition metal cation complexes (Sc^+^, Ti^+^, Mn^+^, Co^+^, and Ni^+^) with NC^−^ and (H_2_O)_8_ ligands, as well as bare Sc^+^- and Ti^+^-catalyst systems to understand the possible roles of such processes in interstellar benzene formation. The results indicate that Sc^+^ and Ti^+^ complexes act as efficient catalysts for cyclotrimerization because they do not have an entrance barrier. The number of d-donor orbitals plays an important role in determining the height of entrance barriers. The NC^−^ ligand reduced the barrier for formation of the Ti^+^ compound. On-the-fly BOMD trajectories were subsequently calculated for TiNC and Ti^+^(H_2_O)_8_ to understand the mechanism of benzene formation from a dynamic perspective. Each trajectory calculation was started from the C_2_H_2_ + TiNC-(C_2_H_2_)_2_ or C_2_H_2_ + Ti^+^(H_2_O)_8_-(C_2_H_2_)_2_ reactants, where the initial distance between C_2_H_2_ and Ti^+^ was set as 4–5 Å. No initial kinetic energy was applied to the atoms in the reaction system to qualitatively understand the overall reaction pathway and the reaction under low-temperature interstellar conditions. BOMD calculations revealed benzene formation via a sequential C-C bond formation in both complexes. Thus, this computational study suggests that interstellar benzene can be formed via acetylene cyclotrimerization catalyzed by a Ti^+^ complex with a solid water (ice) cluster and NC^−^.

It is important to describe the astrophysical implications of the computational study. The most important outcome of the present quantum chemistry calculations is that the benzene formation process starting from C_2_H_2_ + (C_2_H_2_)_2_-Ti^+^-L and C_2_H_2_ + (C_2_H_2_)_2_-Sc^+^-L is barrierless. Nevertheless, it is unlikely that complexes of such metal cyanides with acetylene molecules exist in low-density spaces because of the lack of a third body for absorbing the excess energy. In contrast with isolated ligands, it is likely that the solid surface, such as water and silicate, acts as a ligand for the metal cations [56]. Once a metal cation attaches to the metal center, this could lead to efficient cyclotrimerization. Note that this scenario works only for low-temperature solid water conditions because the metal cation-attached water surface structure is energetically metastable. It should be emphasized that the metal cation can be internally solvated by water molecules owing to the strong ion–water attractive interaction. In fact, many previous experimental studies on large metal cation–(H_2_O)*_n_* complexes have shown that the metal cation is surrounded by several water molecules [57]. Further studies should be performed to further evaluate the quantitative astrophysical implications of the present barrierless cyclotrimerization processes on the dust surface.

In the near future, we will extend the present computational study to include HCN/HNC instead of acetylene in the catalytic cyclotrimerization processes as HCN and its isomer, HNC, are isoelectronic with C_2_H_2_ and have been detected in interstellar media [58]. We intend to discuss the possible roles of transition metal compounds in the formation of pyridine, pyrazine, and pyrimidine via catalytic cyclotrimerization processes, where these compounds play an important role as building blocks in interstellar PAH formation, including heteroatoms. Moreover, CO is abundant in the interstellar medium. We hope to study and discuss PAH formation using CO as the ligand.

## 4. Computational Procedure

Following our previous benchmark quantum chemistry calculations for open-shell transition metal atom systems, [26,27,28] we carried out an unrestricted version of the B3LYP DFT functional with def2-SVPP and def2-TZVP basis sets. To account for the dispersion effects in the B3LYP functional, Grimme’s D3 dispersion corrections (DFT-D3) and the Becke–Johnson (BJ) damping function were employed throughout the present calculations [59]. The potential energies at the stationary points were obtained using the Gaussian09 program package [60]. For reliable computational outcomes, that is, smooth potential energy surfaces for the transition metal compounds with open-shell electronic structures, we applied the “stable = opt” option implemented in the Gaussian09 code at every step of geometry optimization. When this option is selected, the self-consistent field (SCF) solution is forced to converge to the most stable SCF solution without internal instability. All stationary points and their associated IRC were obtained from global reaction route mapping (GRRM) calculations, which automatically explored the potential minima and transition states without human intuition [61,62,63]. The IRC as well as the potential energies at the stationary points calculated by this DFT-D3 level qualitatively agreed with the results obtained by higher-level computational methods. The benchmarked results are represented in the Appendix A. We also performed Born–Oppenheimer molecular dynamics (BOMD) calculations implemented in the Gaussian09 code, where energy gradients were employed to solve the classical mechanical equations of motion. BOMD calculations were performed for TiNC(C_2_H_2_)_2_ and Ti^+^(H_2_O)_8_(C_2_H_2_)_2_ using isolated acetylene systems. Initially, the isolated acetylene was placed approximately 4 Å away from the titanium ion at a constant velocity. The momenta of the TiNC(C_2_H_2_)_2_ and Ti^+^(H_2_O)_8_(C_2_H_2_)_2_ complexes were set as 0. The cation–molecule interactions can affect the activation of nuclear motion.

## Figures and Tables

**Figure 1 molecules-28-07454-f001:**
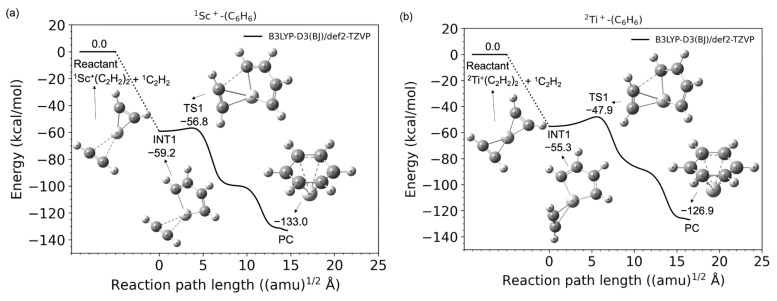
Potential energy profiles for the acetylene cyclotrimerization reactions catalyzed by (**a**) Sc^+^ (singlet spin) and (**b**) Ti^+^ (doublet spin) compounds as a function of the reaction path length obtained at the B3LYP-D3(BJ)/def2-TZVP DFT-D3 level. The zero energy is defined as the energy level of the C_2_H_2_ + TM^+^-(C_2_H_2_)_2_ reactants (TM = Sc and Ti).

**Figure 2 molecules-28-07454-f002:**
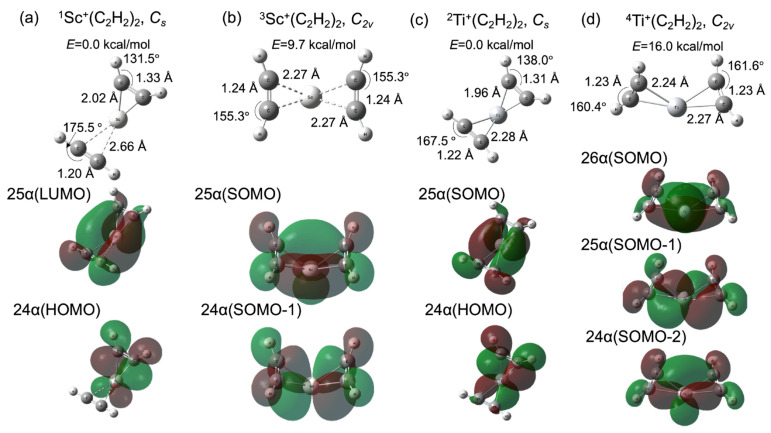
Optimized geometries (**upper** panel) and molecular orbitals (**lower** panel) contributed to π-back-donation for Sc^+^(C_2_H_2_)_2_ of (**a**) singlet and (**b**) triplet states, (**b**) Ti^+^(C_2_H_2_)_2_ of (**c**) doublet and (**d**) quartet (red). The potential energies for the reactants are set as 0 kcal/mol.

**Figure 3 molecules-28-07454-f003:**
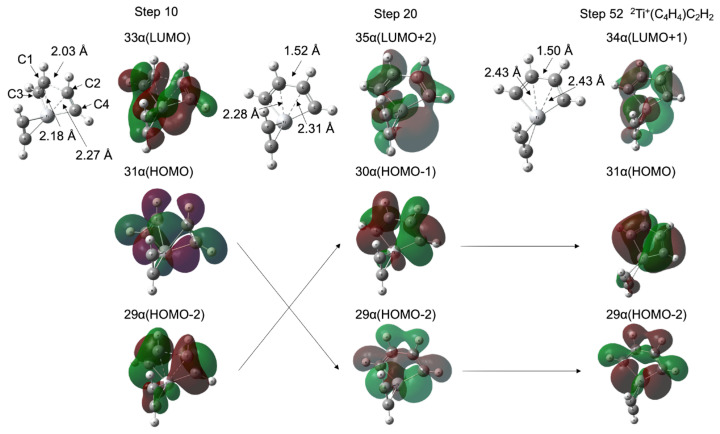
Molecular structures and molecular orbitals at optimization steps 10, 20, and 52 for ^2^Ti^+^-catalyst system. The entire potential energy minimization profiles are shown in Appendix A.

**Figure 4 molecules-28-07454-f004:**
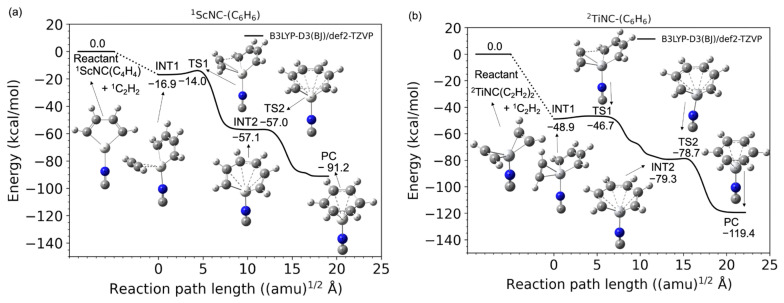
Potential energy profiles for the acetylene cyclotrimerization reactions catalyzed by (**a**) Sc^+^NC^−^ (singlet spin) and (**b**) Ti^+^NC^−^ (doublet spin) compounds as a function of the reaction path length obtained at the B3LYP-D3(BJ)/def2-TZVP DFT-D3 level. Zero energy is defined as the energy level of the C_2_H_2_ + TM^+^NC^−^-(C_2_H_2_)_2_ reactants (TM = Sc and Ti).

**Figure 5 molecules-28-07454-f005:**
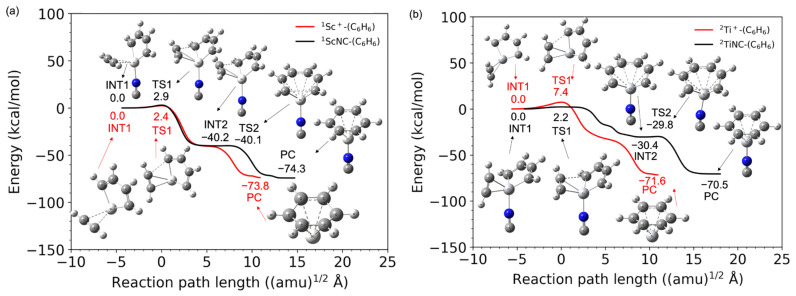
Relative energies as a function of the reaction path length for (**a**) scandium and (**b**) titanium compounds. The red and black lines represent the potential energies for the bare TM^+^ catalyst and TMNC catalyst systems, respectively (TM = Sc and Ti). Zero energy is defined as the energy level of each intermediate complex, INT1.

**Figure 6 molecules-28-07454-f006:**
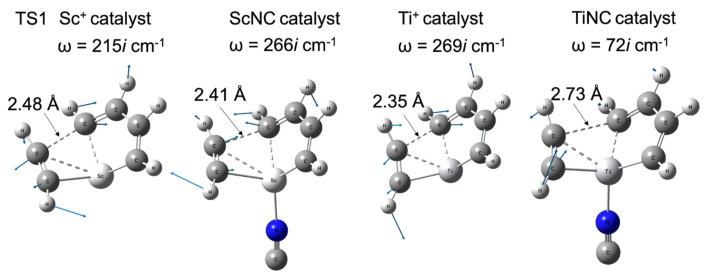
Imaginary frequencies, their vibrational vectors, and important CC distances at TS1 with their structures for Sc^+^, ScNC, Ti^+^, and TiNC catalyst systems.

**Figure 7 molecules-28-07454-f007:**
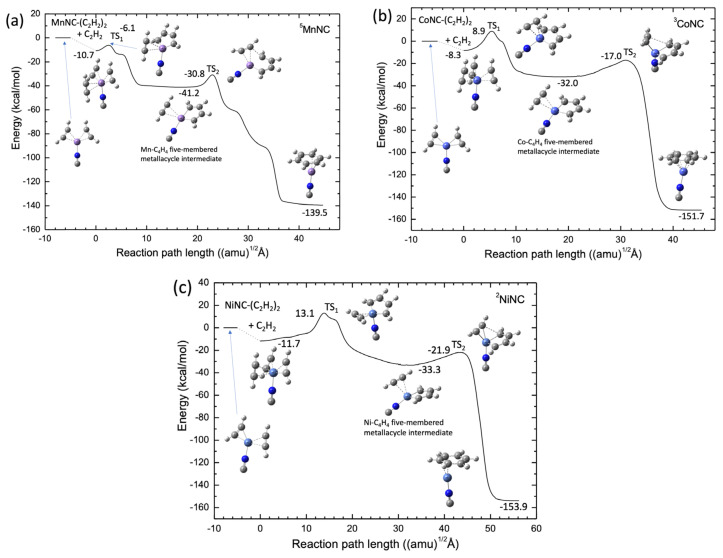
Potential energy profiles for (**a**) Mn^+^NC^−^ (quintet spin), (**b**) Co^+^NC^−^ (triplet spin), and (**c**) Ni^+^NC^−^ (doublet spin) obtained at the B3LYP-D3(BJ)/def2-SVPP DFT-D3 level.

**Figure 8 molecules-28-07454-f008:**
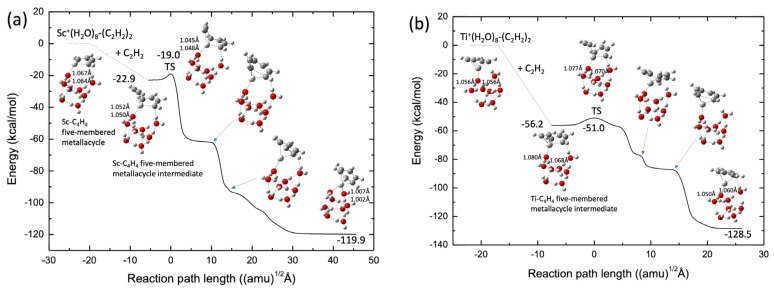
Potential energy profiles for (**a**) Sc^+^(H_2_O)_8_ (singlet) and (**b**) Ti^+^(H_2_O)_8_ (doublet). OH bond lengths (in Å) in water directly bound to the transition metal cation are shown.

**Figure 9 molecules-28-07454-f009:**
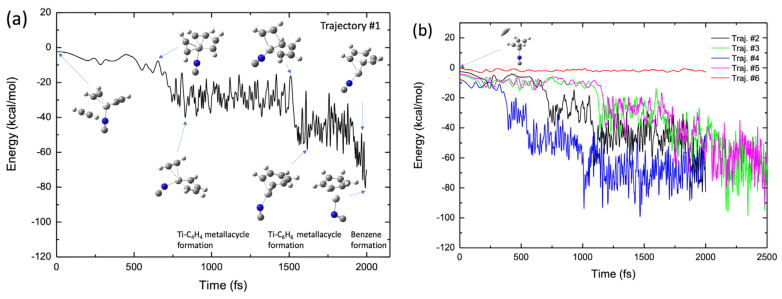
(**a**) Potential energy profile of the benzene formation trajectory for the (C_2_H_2_)_3_-TiNC system plotted as a function of simulation time. (**b**) Potential energy profiles of five trajectories, where four trajectories show benzene formation while one shows non-reactive behavior (see main text).

**Figure 10 molecules-28-07454-f010:**
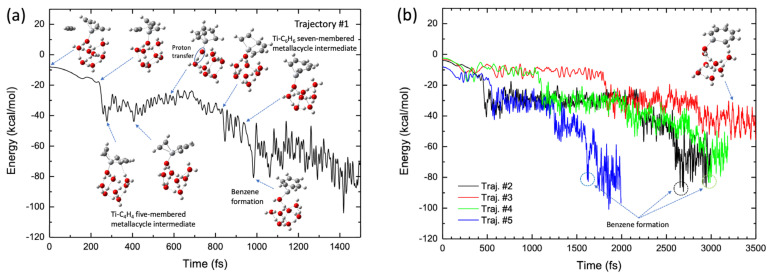
(**a**) Potential energy profile of the benzene formation trajectory for the (C_2_H_2_)_3_-Ti^+^(H_2_O)_8_ system plotted as a function of simulation time. (**b**) Potential energy profiles of four trajectories, where one trajectory shows benzene formation while three trajectories show the five-membered metallacycle intermediate formation.

**Table 1 molecules-28-07454-t001:** Vertical excitation energies (in kcal/mol) at the stationary points for Sc^+^(C_6_H_6_) and Ti^+^(C_6_H_6_) calculated using B3LYP-D3(BJ)/def2-TZVP.

	Reactant	INT1	TS1	PC	Product
Sc^+^-C_6_H_6_					
Quintet	130.5	80.3	79.5	−24.9	549.9
Triplet	36.6	5.3	6.8	−118.7	−79.5
Singlet	0.0	−59.2	−56.8	−133.0	−41.8
Ti^+^-C_6_H_6_					
Sextet	151.3	81.4	79.2	−12.5	626.1
Quartet	40.6	2.8	5.0	−90.5	−74.3
Doublet	0.0	−55.3	−47.9	−126.9	−55.3

## Data Availability

Data are contained within the article.

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
