# Peer review of "Possible Roles of Transition Metal Cations in the Formation of Interstellar Benzene via Catalytic Acetylene Cyclotrimerization"

_molecules, 2023, doi:10.3390/molecules28217454_

Round 1

Reviewer 1 Report

Comments and Suggestions for Authors

In this paper, the authors studied computationally the possible roles of transition metal cations in the formation of interstellar benzene via catalytic acetylene cyclotrimerization. Many similar computational papers have been published, so the novelty of this paper is not good. Considering that the paper is well written and rather comprehensive, publication in Molecules can be recommended after attention to the issues mentioned below:

(1) The routine computations employed by the authors do not relate directly to experiments. There are so many experimental studies on this topic. The authors should combine the experiments such as the infrared spectroscopy of the complexes with their quantum chemical calculations.

(2) Throughout the paper, “B3LYP(D3BJ)” should be “B3LYP-D3(BJ)” and “LC-ωPBE(D3BJ)” should be “LC-ωPBE-D3(BJ)”. In some places, “DFT” should be “DFT-D3”.

Reviewer 2 Report

Comments and Suggestions for Authors

Comments on the Quality of English Language

Reviewer 3 Report

Comments and Suggestions for Authors

From the begining of the manuscript (MS) it is not clear why the authors studied the high-spin states together with the low-spin ones. Later on, they mentioned that the two half-occupied orbitals contribute to the π-back donation in the Sc+(4s3d) (C2H2)2 complex. Very detailed, but poorely explained Figures 2-4 and 9, seem to clarify the importance of triplet and quintet states involvement into calculation of coordination reaction paths. In this respect, some references on the spin uncoupling mechanisms in the 3d-metal - acetylene interaction and coordination are necessary. In the Ref. [1]  Spin uncoupling in surface chemisorption of unsaturated hydrocarbons. L Triguero, LGM Pettersson, B Minaev, H Ågren. The Journal of chemical physics 108 (3), 1193-1205 (1998) the importance of triplet state configurations during metal  coordination to alkynes was outlined and connected with "direct and back-donation" D-A mechanisms.  It was shown in [1] that unsaturated hydrocarbons, such as acetylene, benzene and ethylene, show strong geometrical distortions when coordinated to transition metals or to surfaces; the bonding is normally analysed in terms of a π-donation—-backdonation process. In that work the chemisorption of the unsaturated hydrocarbons (ethylene, acetylene, and benzene) on cluster models of the copper surfaces was used to demonstrate the importance of considering the available excited states of the free molecule in analyzing the bonding scheme of the adsorbate at the surface. By comparison to the structures of the triplet excited states in the gas phase it was demonstrate that these must be considered as the states actually involved in the bonding. This implies a spin-uncoupling in both adsorbate and substrate as part of the chemisorption process or metal-organic bond formation.

[2] Thermally accessible triplet state of π-nucleophiles does exist. Evidence from first principles study of ethylene interaction with copper species. SV Bondarchuk, BF Minaev. RSC Advances 5 (15), 11558-11569 (2015)

These references are important from the historical perspective and the fair description of the problems (Are the methods adequately described? Are the results clearly presented?)

Interpretation of numerous MOs is not clear. The number of MOs and Figures could be restricted with the clear discussion of the bonding-antibonding features in connection with the metal catalysis mechanism and spin-uncoupling problems. 

This MS belong to a series of similar papers [21-23]. The role of dynamic simulation and reaction coordinate should be explained in more clear fashion. The MS presents interesting results which are obtained by adequate and rather sophisticated approximations. The MS can be accepted after minor revision.

Round 2

Reviewer 2 Report

Comments and Suggestions for Authors

The authors have taken care to respond to all my comments, which I think makes the article easier to read. I therefore recommend publication of this work in Molecule.

Comments on the Quality of English Language

No comments.

Reviewer 3 Report

Comments and Suggestions for Authors

The manuscript about  the roles of metal ions in the benzene formation via Catalytic Acetylene Cyclotrimerization (CAC) is interesting from the point of fundamental theory of metal catalysis and from the astrochemical observations importance. Very detailed DFT calculations are presented for the Sc, Ti, Mn, Co, and Ni ions as catalysts in CAC reactions. In the revised version the recommended propositions are taken into account. The role of spin uncoupling in the coordination process is discussed and the relevant references are added. The MS is improved and can be recommended for publication.